# Assessment of Herd, Calf, and Colostrum Management Practices on Austrian Dairy Farms Using a Scoring System

**DOI:** 10.3390/ani13172758

**Published:** 2023-08-30

**Authors:** Nicole Hechenberger, Katharina Lichtmannsperger, Daniela Klein-Jöbstl, Alexander Tichy, Thomas Wittek

**Affiliations:** 1Animal Health Service (Tiergesundheitsdienst) Salzburg, Bundesstraße 6, 5071 Wals-Siezenheim, Austria; nicole.hechenberger@salzburg.gv.at; 2Clinical Unit of Ruminant Medicine, University Clinic for Ruminants, Department for Farm Animals and Veterinary Public Health, University of Veterinary Medicine Vienna, Veterinärplatz 1, 1210 Vienna, Austria; thomas.wittek@vetmeduni.ac.at; 3Clinical Unit for Herd Health Management in Ruminants, University Clinic for Ruminants, Department for Farm Animals and Veterinary Public Health, University of Veterinary Medicine Vienna, Veterinärplatz 1, 1210 Vienna, Austria; daniela.klein@vetmeduni.ac.at; 4Bioinformatics and Biostatistics Unit, University of Veterinary Medicine Vienna, Veterinärplatz 1, 1210 Vienna, Austria; alexander.tichy@vetmeduni.ac.at

**Keywords:** calf management, herd management, survey, colostrum

## Abstract

**Simple Summary:**

As calves are born with a naïve immune system, they depend on the transfer of immunoglobulins via colostrum. If calves do not receive a sufficient supply of high-quality colostrum (>50 g/L of immunoglobulins; 100–300 g Immunoglobulins in total) within the first hours after birth, they are likely to suffer from Failure of Transfer of Passive Immunity. The objective of the present study was to evaluate herd, calf, and colostrum management on Austrian dairy farms, focusing on challenges and possibilities for improvement. A scoring system was implemented to compare and classify management practices. Farms in foothills/flatland regions of Austria, conventional producing farms, and full-time operated farms overall received a higher and therefore better score rating than farms in alpine regions of Austria, organic producing farms, and part-time operated farms.

**Abstract:**

The objectives of the study were to describe colostrum management on Austrian dairy farms and to explore differences between regions (alpine/flatlands), organic and conventional producing farms, and full-time or part-time operated farms. An online survey (24 questions) on general farm characteristics and herd and calf management was sent to 16,246 farmers. In total, 2328 farmers (response rate 14.3%) answered the questionnaire. To allow an objective comparison, a scoring system was implemented. Farm size is, on average, smaller in the alpine regions than in the foothills/flatlands regions of Austria. Small farms were more often organic-producing farms (81.6%) and operated part-time (93.8%). In foothills/flatland regions, 70.0% of farms have a separate calving area, and in the alpine regions, it is solely 42.8%. Colostrum testing is still mostly done by visual appraisal (63.7%); only a few farmers use a colostrometer (8.8%), brix-refractometer (18.3%), or ColostroCheck^®^ (9.2%, a cone-shaped device to rate the flow velocity of colostrum). The results of the present study using the scoring system showed differences in herd and calf management practices in all sectors. In the future, the findings and especially the scoring system can support Austrian dairy farmers or veterinarians to better assess areas of improvement on farms in order to prevent calves from suffering from Failure of Transfer of Passive Immunity.

## 1. Introduction

Cattle have an epitheliochorial placenta type. The maternal uterine tissue layers remain intact, resulting in a separation between the maternal and the fetal blood circulation [1,2]. Therefore, an intrauterine transfer of immunoglobulins is almost impossible, and calves are born with a naïve immune system [3]. Unless an adequate amount of immunoglobulins is provided via colostrum, calves have an increased likelihood of succumbing to infections [4]. The process of transferring maternal immunoglobulins via colostrum to the calf is called Transfer of Passive Immunity (TPI). The amount of maternal colostrum needed depends on the concentration of immunoglobulins, the ingested volume of colostrum, and the ability of the calf’s gut to absorb the immunoglobulins (apparent efficiency of immunoglobulin absorption). If calves are fed with >2.5 L colostrum (>50 g/L IgG) within the first three hours after birth, they are less likely to suffer from FTPI [5]. During the first twelve hours after birth, the calf’s ability to absorb maternal immunoglobulins decreases substantially [6]. The reasons for this are not yet fully understood, and some findings are conflicting [7]. Some studies showed the presence of a tubular vesicle-vacuolar mechanism in neonatal enterocytes. The vacuoles that transport the immunoglobulins from the intestine to the blood decrease over time as the fetal intestinal cells mature [8]. The insufficient supply of the calf with immunoglobulins via a low amount of immunoglobulins in maternal colostrum or insufficient amount of colostrum is termed a Failure of Transfer of Passive Immunity (FTPI). Reschke and coworkers (2017) investigated 373 mother-dam pairs, where 162 (43.50%) of the calves showed an FTPI [9]. A sufficient maternal colostrum quality is not only defined by the amount of immunoglobulins but also by the level of bacterial contamination [10]. Colostral immunoglobulins, which are bound to bacteria in the colostrum, are deactivated before they can be absorbed [11]. Furthermore, bacteria present in colostrum compete with immunoglobulins on the unspecific receptors, which are necessary for absorbing immunoglobulins from the intestine into the bloodstream, resulting in a decreased apparent efficiency of immunoglobulin absorption [12]. It is essential to use adequate management methods for collecting, storing, and feeding colostrum to ensure calves get an appropriate amount of colostral immunoglobulins after birth and, in consequence, do not suffer from FTPI [13]. Additionally, cow-related factors such as the number of lactations, genetic parameters, dry period length, antepartum milk leakage, colostrum quantity, metabolic status of the cow, and udder health can influence colostrum quality [9,14,15,16,17,18]. Furthermore, colostrum and nutritional management of calves (quantities, occurrence of FTPI, etc.) have an effect on the future adult dairy cow (epigenetic programming) [19,20,21]. Calves suffering from FTPI have higher mortality and morbidity rates and reduced daily growth rates [20]. They are more likely to suffer from diseases such as diarrhea, respiratory diseases, navel infections, and omphalitis [13,22,23,24]. Calves suffering from FTPI result in substantial economic losses for farmers, and additionally, FTPI poses a major animal welfare issue [25].

In 2012, calf management practices have been evaluated in Austria [26]. The study described calf management practices, estimated differences in disease incidences on Austrian dairy farms depending on the farm structure (small farms ≤ 20 cows versus large farms > 20 cows), and different management practices [18]. Investigations carried out in countries such as the Netherlands [27], Brazil [28], Canada [29], and the USA [30] are only partially comparable to studies conducted in Austria since the farm structure is significantly different. In the aforementioned studies, the average number of cattle per farm is significantly higher, with more than 50 to 100 dairy cows per farm. In Austria, there are primarily small family-owned farms with an average size of 19 dairy cows per dairy farm [31]. This structure leads to the fact that many farms are run as part-time farms [32]. They are similarly structured like full-time farms. The only difference is that the farm owners have an additional income besides dairy farming (e.g., from agritourism, employment, wage work, and forestry). The on-farm produced raw milk is either collected using a dairy company and further processed in the dairy plant and/or processed directly on the dairy farm and sold as cheese and/or milk at the local farmer’s store. Due to the different geographical structures, Austria has diverse dairy farming structures [32]. Cows calf all year round, but especially in the alpine region, block calving is conducted, and the cows’ calves from around September to January. This is carried out due to the fact that alpine transhumance is carried out, which means that the dairy farmers with their respective cows and young stock move to mountainous areas during summertime. Additionally, Austria has a high number of organic-producing farms, especially in Salzburg, and therefore, the cows are on the pasture from May until October. In the western and southwestern parts of Austria, there are mostly small farms with less than 30 dairy cows per farm due to the mountain range of the Austrian Alps. In the southeast and eastern parts of Austria (foothills and flatlands), the farms tend to be larger, with ≥30 dairy cows per farm. For details on the geographical differences, see Figure 1.

The objectives of this study were to describe herd and calf management practices in Austria and describe differences in colostrum management between (1) alpine (west) and foothills/flatland (east) regions, (2) between organic and conventional production types and (3) between part-time and full-time operated farms by implementing a scoring system.

We hypothesized that there were differences in herd and calf management practices between (1) farms located in the alpine (west) or foothills/flatlands (east) regions of Austria, (2) between organic and conventional farms, and (3) between part-time and full-time operated farms.

## 2. Materials and Methods

### 2.1. Survey Distribution

An online survey was designed using Survey Monkey^®^ (Copyright © 1999–2022 Momentive). The survey was distributed to 16,246 dairy farmers in Austria who were members of the Austrian Breeding Association (ZAR, Rinderzucht Austria) via email using their member database. The study population of Austrian breeding association members covers 68.1% of the total dairy farms in Austria. The survey was open for 8 weeks, beginning 1 February 2022 and ending 31 March 2022, with a reminder to participate in the survey sent out at the beginning of March 2022.

### 2.2. Survey Structure

In total, the survey included 24 questions. All 24 questions were single-choice questions, some with the possibility to provide an additional open answer. Additional supplementary questions were possible for three questions, depending on the answers given (‘If yes, …’). Overall, the questions were divided into three sections: Section (1) general farm characteristics, Section (2) information on herd-management procedures, and Section (3) information on calf-management procedures. Section one included questions on the location of the farm (federal state), Austrian Animal Health Service membership (yes/no), farm size (in terms of livestock units and number of cows), production type (organic/conventional), operation type (full-time/part-time), housing types (tie stall/free stall) and cattle breeds. Within the second Section, more specific information on herd management practices with a special focus on herd-level colostrum management was gathered, such as the availability of a calving area, udder cleaning methods, duration between parturition and colostrum milking and colostrum storage procedure. In the third Section, detailed information on calf management practices was gathered, including questions on colostrum feeding procedure (bucket feeding, nipple bottle feeding), feeding time, and colostrum quality assessment methods. The original survey (in German and English) has been provided in the Appendix A.

### 2.3. Implementation of a Scoring System

In order to compare herd and calf management practices, a scoring system was implemented for 13 questions (Section 2 and Section 3). A high score agrees with evidence-based recommendations on herd and calf management procedures. A low score indicates that the answers were not in accordance with evidence-based recommendations. Each answer was translated into a point system (minimum point = 0 points; maximum points = 4 points). In total, a maximum of 32 points could be obtained per farm: 14 points for herd management practices and 18 points for calf management practices. The point allocation was based on the current evidence-based recommendations published in peer-reviewed journals. In detail, the information from published peer-reviewed articles was used as a basis for the discussion round by the authors (NH, KL, TW). Answers according to the current evidence-based recommendations were assessed as ‘correct’ and received higher scores (3 and 4 points, mostly or fully meets the evidence-based recommendation) than ‘incorrect’ answers (1 and 2 points, does not or only partially meet evidence-based recommendations). Questions solely ending in a yes/no decision were categorized as a correct (1 point) and an incorrect answer (0 points). If superiority to a specific method/procedure over another could not be determined, an equal number of points was given. For example, Section Two included the information on the udder cleaning method, and two answers were possible (yes/no). The answer “yes—the udder was cleaned before colostrum harvest” resulted in one point, and the answer “no—the udder was not cleaned before colostrum harvest” resulted in zero points. The allocation of points was based on the publication by S. Steward et al. 2005 [33], where the authors found that a high standard of udder cleaning is essential to harvest high-quality colostrum with low bacterial counts. Our survey did not ask about the detailed udder cleaning routine. Therefore, the question about the cleaning method was excluded from the scoring system, and only the question cleaning ‘yes/no’ was scored. Figure 2 provides two more sample questions to illustrate the allocation of points within the scoring system.

The complete survey, including the point allocation system (including the references of the current evidence-based recommendations), has been provided in the Appendix A.

### 2.4. Extracting Data from Two Official Databases

Two main Austrian databases were used to gather data regarding farm structure, production type, and Animal Health Service membership. The first database which was used is owned by the AgrarMarktAustria (AMA). This institution was established in 1992 by law and is a subject of the Bundesministerium für Land- und Forstwirtschaft (Federal Ministry of Agriculture and Forestry). The AMA is authorized to report on national and international agriculture markets and advancements in agriculture. The second database used was the Verbrauchergesundheitsinformationssystem (VIS), which is a governmental database operated for the Bundesministerium für Soziales, Gesundheit, Pflege und Konsumentenschutz (Federal Ministry of Social Affairs, Health, Care and Consumer Protection). All livestock farms and private establishments keeping animals (Zoos, equine holdings, etc.) are required by law to be registered in the VIS.

### 2.5. Statistical Analysis

The answers to the internet-based survey were transferred to Microsoft Excel 2016 (Microsoft Office Professional Plus 2016 © Microsoft, Redmond, WA, USA). All answers to the survey were viewed individually for plausibility and contradiction; not plausible and contradictory answers were removed from the statistical analysis. The federal state of Vienna, which is the capital city of Austria, was excluded from the study since there is no commercial dairy farming. The survey was distributed throughout Austria via the member database of the Austrian breeding association, and the farmers participated on a voluntary basis. In order to quantify the representativity of the survey, the official Austrian databases (Section 2.4) were used to compare the survey results to the overall Austrian population using descriptive statistics (% survey response versus % overall Austria). The single-choice questions were coded, and the open answers were assessed individually and categorized if suited or not taken into statistical account. The data were described using descriptive statistics expressed as median, 10, 25, 75, and 90 percentiles, minimum and maximum values. The number of dairy cows on the farm (Ncow) and the average 305-day milk yield were categorized. The number of dairy cows on the farm was categorized as follows: ≤10 dairy cows, 11 to 20 dairy cows, 21 to 30 dairy cows, 31 to 40 dairy cows, and ≥41 dairy cows. The average 305-day milk yield was categorized as follows: no or implausible information available; 2000 to 6500 L; 6501 to 7500 L; 7501 to 8700, and 8701 to 14,000 L. If there was no information (missing values) on the location of the farm (federal state), Animal Health Service membership, production type (organic/conventional farming), and/or operation type (full-time/part-time), the questionnaire was excluded from further statistical analysis.

The final data file was transferred to SPSS^®^ statistics software Version 28 (IBM^®^, New York, NY, USA) for further investigations. The herd management score and the calf management score were tested for normality using the Kolmogorov–Smirnov test, including the Lilliefors correction. The data were not normally distributed (*p* < 0.05). Therefore, the non-parametric Mann–Whitney U test (two independent variables) and the Kruskal–Wallis test (>2 independent variables including the Bonferroni correction) was applied to test if there are statistically significant differences between herd management and calf management practices between organic and conventional farms, part-time and full-time farms and between the different federal states of Austria. According to similar geological structures, the federal states of Austria were divided into alpine regions (high alpine and alpine uplands) in the west, including Vorarlberg, Tyrol, Salzburg, and Carinthia and foothills and flatland/hill country regions in the east, including Upper and Lower Austria, Styria and Burgenland. The level of significance was set at *p* < 0.05.

## 3. Results

### 3.1. Response Rate

A response rate of 14.3% (16,246 surveys sent/2328 answers returned) was calculated. In total, 72/2328 (3.1%) answers had to be removed due to missing values for the questions of the federal state, Animal Health Service, production type, and operating type. All the answers given by cow-calf operations (46/2328; 2.0%) were removed since the number was too low for a meaningful analysis. After removing these 118 answers, a total of 2210 answers (94.9% of all answered surveys; response rate: 13.6%) were used for the final statistical analysis. A detailed overview of the overall number of dairy farms in Austria, the production type (organic/conventional), operation type (part-time/full-time), and the Animal Health Service Membership (Tiergesundheitsdienst, TGD), including the survey results are shown in Table 1.

### 3.2. General Farm Characteristics

The results of this survey section on general farm characteristics were split. Regarding the farm size, Tyrol had most farms in the category of less than 10 dairy cows per farm (43.6% of all answers from Tyrol). Vorarlberg (32.5%), Salzburg (36.4%), and Carinthia (41.1%) showed most of the farms within the category of 11 to 20 dairy cows per farm. In Styria (29.6%) and Lower Austria (28.5%), the majority of farms were within the category of 21 to 30 dairy cows per farm. The largest farms were located in Upper Austria (31.4%) and Burgenland (75.0%), within the category of more or equal to 41 dairy cows per farm.

Simmental is the major breed in all federal states (n = 1636, 74.3%), Brown Swiss/Original Brown (n = 287, 13.0%) primarily in Tyrol (24.9%) and Vorarlberg (67.5%), Holstein-Fresian 7.1% (n = 156), Pinzgauer (n = 65, 3.0%) primarily in Salzburg (14.3%) and Tyrolean Grey (n = 37, 1.7%) primarily in Tyrol (7.6%). All other breeds (Jersey, Tuxer, Belgian Blue, etc.) and crossbreeds were summarized as one category (n = 22, 1.0%).

Regarding the average milk yield in Salzburg (35.8%) and Tyrol (33.5%), most respondents were within the category of producing 2000 to 6000 L per cow per year. In Vorarlberg, most respondents were in the category of 7501 to 8700 L (30.8%). In Burgenland, Styria, Upper and Lower Austria, most respondents were in the category of 8701 to 14,000 L (62.5%, 32.3%, 31.0%, 35.1%, respectively). In Carinthia, there were equal answers (24.5%) in categories 6501 to 7500 and 8701 to 14,000 L per cow per year.

In Burgenland (50.0%), Carinthia (49.7%), Salzburg (49.1%), and Styria (47.2%), most farms house their lactating and dry dairy cows in a free-stall barn with an outdoor loafing area (OLA) and/or a pasture. In Upper (42.7%) and Lower Austria (44.1%), a free-stall barn without an OLA and/or pasture is the most common housing type. In Tyrol (68.2%) and Vorarlberg (49.2%), the cows were primarily housed in tie stalls with OLA and/or pasture.

The livestock units were not evaluated due to many missing values (383, 17.33%) and many contradictory answers (219, 9.9%). Therefore, the number of dairy cows on the farm (Ncows) was used to compare farm sizes. The question on the membership of the official national milk performance recording organization produced confusing answers. Obviously, the respondents did not know the official name of the milk performance organization or they state their membership in a breeding association. Consequently, due to multiple contradictory answers, this open question was excluded.

An overview of the results from the survey section one on general farm characteristics is provided in Table 2. The survey results for all federal states of Austria are shown in detail in the Appendix A.

### 3.3. Herd Management Practices

The second Section contained questions on herd management practices on dairy farms with a special focus on herd-level colostrum management practices. The results for alpine (west) and foothills/flatland (east) regions, organic/conventional production types, and part-time/full-time operated farms are provided in Table 3 and Table 4. The total data set on herd management practices in the eight federal states of Austria is provided in the Appendix A.

### 3.4. Calf Management Practices

The third Section contained questions on calf management practices on dairy farms. The results for alpine (west) and foothills/flatland (east) regions, organic/conventional production types, and part-time/full-time operated farms are provided in Table 5 and Table 6. The total data set on calf management practices in the eight federal states of Austria is provided in Appendix A.

#### 3.4.1. Colostrum Quantity

The usual quantity of colostrum within the first six hours is 2–4 L in Burgenland, Carinthia, Lower Austria, Upper Austria, Salzburg, Styria, Tyrol, and Vorarlberg (37.3%, 65.6%, 72.3%, 73.3%, 73.8%, 71.2%, 65.8%, 64.2%). Less than two liters are fed in 25.0% (Burgenland), 22.5% (Carinthia), 19.3% (Lower Austria), 19.3% (Upper Austria), 9.2% (Salzburg), 18.5% (Styria), 21.3% (Tyrol) and 23.3% (Vorarlberg). For detailed results, see Appendix A.

#### 3.4.2. Colostrum Feeding Methods

The nipple bottle is the most reported feeding method in Burgenland, Carinthia, Lower Austria, Upper Austria, and Styria (50.0%, 64.9%, 64.8%, 67.0%, and 65.8%). In Salzburg, Tyrol, and Vorarlberg, it is the bucket (61.3%, 54.6%, and 53.8%). If the calf does not drink colostrum willingly, colostrum is offered multiple times, and in no case, an esophageal tube be used in 42.9% (Burgenland), 62.7% (Carinthia), 66.2% (Lower Austria), 56.6% (Upper Austria), 68.4% (Salzburg), 64.4% (Styria), 71.0% (Tyrol) and 78.2% (Vorarlberg). On the other hand, an esophageal tube is used within 2–6 h in 28.6% (Burgenland), 22.7% (Carinthia), 22.9% (Lower Austria), 26.3% (Upper Austria), 21.0% (Salzburg), 19.1% (Styria), 15.9% (Tyrol) and 18.5% (Vorarlberg). For detailed results, see Appendix A.

### 3.5. Assessing Herd- and Calf Management Practices by the Use of a Scoring System

The overall median for Austria in herd management practices was 9 (range 1–14), and in calf management practices, 12 (range 3–18). Comparisons of herd and calf management scores for alpine (west) and foothills/flatland (east) regions, production types (organic versus conventional), and operating types (full-time versus part-time) are shown in Figure 3 and Figure 4.

#### 3.5.1. Differences between Alpine and Foothills/Flatland Regions

In the alpine regions (west), there was a statistically significant difference in herd management between Tyrol and Salzburg (*p* < 0.01) and Tyrol and Carinthia (*p* < 0.01); for details see Appendix A. In foothills/flatland regions (east), there was no statistically significant difference between each other for herd management. In all of Austria there were statistically significant differences for herd management between Lower Austria and Salzburg (*p* < 0.01), Vorarlberg (*p* < 0.01), and Tyrol (*p* < 0.01); between Upper Austria and Vorarlberg (*p* < 0.01), Salzburg (*p* < 0.01) and Tyrol (*p* < 0.01); between Styria and Vorarlberg (*p* < 0.01), Salzburg (*p* < 0.025) and Tyrol (*p* < 0.01). In calf management, there were only statistically significant differences between Upper Austria and Vorarlberg (*p* < 0.01), Tyrol (*p* < 0.01), and Styria (*p* < 0.01). The details are shown in the Appendix A.

#### 3.5.2. Differences between Organic and Conventional Farms

##### Herd Management Score

Organic and conventional farms had a median score of 9 in herd management practices. Percentiles 10, 25, 75, and 90 for organic-producing farms were 6, 7, 11, and 12, respectively (*p* < 0.01). Percentiles 10, 25, 75, and 90 for conventional farms were 6, 8, 12, and 13, respectively (*p* < 0.01). Organic farms had a larger statistical spread than conventional farms.

##### Calf Management Score

The median for organic and conventional farms was 12. The percentiles 10, 25, 75, and 90 were 9, 11, 12, 13 for organic farms and 10, 11, 13, 14 for conventional farms (*p* < 0.001).

#### 3.5.3. Differences between Part-Time and Full-Time Farms

##### Herd Management Score

Full-time operated farms had a median score of 10 in herd management practices, whereas part-time operated farms had a median score of 8. The percentiles 10, 25, 75, and 90 for full-time operated farms were 6, 8, 12, 13, and for part-time operated farms, 6, 7, 10, 12 (*p* < 0.01).

##### Calf Management Score

For full-time and part-time operated farms, the median was 12 with a 10, 25, 75, and 90 percentile for full-time: 9, 11, 13, 14 and part-time: 10, 11, 12, 13, respectively.

## 4. Discussion

### 4.1. Survey Design and Response Rate

The response rate was reasonable, with 14.3% in the present study. Other similar internet-based, voluntary surveys conducted in Austria showed response rates of 12.2% (1287 respondents) [26] and 11.3% (1018 respondents) [34]. Another similar email-based survey was conducted in Germany on the evaluation of fresh cow management with a response rate of 12.0% (429 respondents) [35]. Due to the fact that all studies carried out convenience sampling, the external validity is limited. The study was not designed as a representative survey. Still, the survey provides a good overview of the herd and calf-level colostrum management practices in Austria since the number of farms per region (federal states) was comparable to the relative number of farms per federal state, as provided in Table 1. The responses of organic-producing farms were, in total, overrepresented as only 27.5% of dairy farms in Austria are organically producing, and in the present study, 30.7% were organic-producing farms. For details, see Table 1. The number of farms with an Animal Health Service Membership was higher in the present study (96.7%) than it actually is in Austria (91.5%, see Table 1). It can be hypothesized that more educated and motivated farmers participated in the survey. The Animal Health Service operates in all eight federal states of Austria (except Vienna, which is the capital city of Austria but also a federal state) and is an association that farmers can join on a voluntary basis. The Animal Health Service membership includes regular farm visits from veterinarians, including herd health checks, discussing management procedures, and biosecurity measurements. Members also have to fulfill a certain number of continuing education courses throughout the year. In the federal state of Vorarlberg, the Animal Health Service was mandatory for all cattle farmers in 2022.

### 4.2. Scoring System

The scoring system, including the allocation of points, was implemented in order to have a tool to objectively compare the different herd management and calf management practices. The authors are aware of the fact that scientific opinion is on the lowest level on the pyramid of evidence [36]. The authors based their opinion on articles published in peer-reviewed journals. When the farmers receive the results from the scoring system, they have the possibility to focus specifically on the weaknesses in colostrum management on their farms. In the future, the scoring system might be used to predict the likelihood of FTPI in calves on farms, as similar scoring systems have been implemented in human medicine for decades to predict disease (e.g., prediction of coronary heart diseases [37]). Further studies will be needed. Currently, it is not possible to see if calves are at risk of facing FTPI since we do not have data from the calves. At this stage, the scoring system should rather be used to educate farmers regarding current evidence-based recommendations. Three questions were not included in the scoring system due to the fact that there was no conclusive research available when the study was conducted. Therefore, no clear distinction between the methods was feasible (see Appendix A). Additionally, our scientific panel consisted of three panelists. In future studies, it is recommended to set up a panel with more scientists discussing the point allocation and/or collect actual data on FTPI in calves on each farm.

Our results show higher scores for the herd (*p* < 0.01) and calf management (*p* = 0.02) practices in foothills/flatlands regions. The possible reasons for this might be more full-time operated farms, the availability of land to build new facilities due to the low population density, and/or more conventional producing farms. Calf management practices seem similar in all federal states of Austria and do not differ much in the different production and operating types, although all findings were statistically significant (*p* < 0.01). Although the median score was the same (12) for all regions, production, and operating types, alpine regions, organic producing, and part-time operated farms have a bigger statistical spread of answers. It seems that the knowledge of the importance of fast colostrum administration and high colostrum quantities is widely known in Austria and does not depend on the region, production, or operating type. Our study showed that there are small farms with a high knowledge of good colostrum management but also large farms with an insufficient knowledge of good colostrum management in Austria. In the future, the scoring system might be used by farms to find points of improvement for their colostrum management practices and to implement a guideline to standard operating procedures (SOP) for small-scale dairy farms in Austria as has been carried out in Germany [38].

### 4.3. General Farm Characteristics

Farm size in Austria (19 dairy cows [31]) is in accordance with the present study (28.7% 11–20 dairy cows per farm). In Austria, there is a large difference between farms in alpine and foothills/flatland regions. In Tyrol, an average of 21.6 cattle are kept per farm, whereas in Burgenland, on average, 48.9 cattle are kept on farms [39]. This might lead to more part-time operated farms in alpine regions. Additionally, agricultural tourism plays a significant role in the alpine regions of Austria. The distribution of major cattle breeds in Austria is Simmental 74.7%, Holstein-Fresian 7.3%, Brown Swiss 5.7%, Pinzgauer 2.0%, Tyrolean Grey 0.9%, which is very similar to the present study (see Table 2). Most Brown Swiss and Tyrolean Grey are on farms in Tyrol and Vorarlberg [40], also similar to our study.

The average milk yield was lower in alpine (most respondents within the category of producing 2000 to 6000 L per cow per year) than in foothills/flatland regions (category 8701 to 14,000 L per cow per year). This is probably because there are more organic producing and part-time operated farms in alpine regions. Those farm types seem, according to our study, to be more extensive, producing farms with lower milk yields and fewer dairy cows.

Barn types differ in alpine and foothills/flatland regions. Alpine regions might run tie-stall barns more often due to the farm size of ≤30 dairy cows. Additionally, in the alpine regions, there are plenty of part-time farms where the farmers gain additional income besides farming.

### 4.4. Herd Management Practices

The results indicate that it is more common for organic farms (18.7%) to leave the calf with the dam for more than 4 h than it is for conventional farms (9.2%). If the calf is left with the dam, there is no way of knowing in what time frame and how much quantity of colostrum the calf consumed. Furthermore, it can be assumed that those farmers also do not test colostrum quality. This practice might be due to the philosophy of organic producing farms (‘close to nature’). However, it is indicated by studies that calves suffer less likely from FTPI when separated from the dam within 3 h [41,42]. Nevertheless, the apparent efficiency of colostrum absorption might be better if the dam is present. Therefore, further studies are needed to verify this effect on the dam and the calf. The results of the present study (Table 3) showed that in the survey, 57.6% of Austrian farms have a separate calving area. In a previous survey conducted in Austria, 47.0% of farms answered that they do have a separate calving area, and 51.1% did not [26]. This shows an improvement in Austrian management practices. The availability of a separate calving area is recommended to ensure high hygiene at birth for the calf and to minimize stress for the cow [41,43]. The present study showed that alpine regions and part-time operated farms frequently do not have a separate calving area. On organic and part-time operated farms, results showed that even if there is a separate calving area available at the farm, in 20.4% (organic farms) and 18.9% (part-time farms), half or even less than half of the cows calve in this area. Our survey did not ask for the reasons why cows do not calve in a separate area if one is available. The reasons might be that in Austria, oftentimes, the separate calving area is also used for diseased animals and, therefore, not available for periparturient cows and not disinfected regularly, which represents a risk factor for spreading diseases [44]. Other reasons might be the time management on part-time operated farms. Another possible explanation might be that in tie-stall barns, cows usually calve in their stall as there is no separate calving area available. A prolonged time period (>6 h) until first milking the dam is associated with a low colostrum quality [9]. Milking the dam at the next standard milking time as a usual procedure may conclude in a longer time period than six hours and, therefore, may result in FTPI in calves. Therefore, this practice as a standard procedure is not recommended. That knowledge seems to be widely known amongst Austrian farmers, although, according to the underlying results, farms from foothills/flatland regions (12.0%) rely on that practice more often than farms in alpine regions (8.4%).

Storing frozen colostrum is recommended by many authors [45,46]. Results from our study show that in foothills/flatland regions, the information on storing frozen colostrum is more well-known than in alpine regions, see Table 3. That is probably because, in alpine regions, there are more part-time operated farms. Additionally, the results showed that part-time-operated farms were less likely to have frozen colostrum storage.

In other studies, critical control points for bacterial contamination of colostrum have been established, one of them being the harvesting process [33]. We assumed that when colostrum is harvested by hand, the udder is not sufficiently cleaned beforehand, but only 13.6% explicitly stated that they do not clean the udder before milking. So, it can be presumed that some of those farms that harvest colostrum by hand also clean the udder beforehand. We also did not ask if they use gloves when milking colostrum by hand. No usage of gloves might again lead to bacterial contamination of the human skin [47]. In the future, this question needs to be altered in order to gain more information on the hygiene practices when harvesting colostrum by hand. Another study comparing total bacterial counts (TBC) in colostrum samples collected directly from the teat and from feeding equipment showed that TBC was higher in samples collected from feeding equipment, suggesting that the focus should lie on the hygiene of colostrum harvest and feeding equipment [48]. It is shown in different studies that dry teat cleaning lowers the bacterial count, and wet cloths should only be used if the teat is dried before milking [49]. Concerning the methods used for udder cleaning, our survey did not ask about the detailed udder cleaning routine. This should be improved in future studies using a scoring system.

### 4.5. Calf and Colostrum Management Practices

High-quality colostrum from the respective dam should always be preferred to pooled colostrum [3,50]. In our study, 66.3% always use colostrum from their own dam. In Austria, it is not common practice to pool colostrum. On small farms, there is often only one cow in parturition at any one time. This particular question may have produced some confusion because 29.1% checked that they do ‘not always’ (1.1%) or only ‘if the dam has good colostrum quality’ (28.0%) use colostrum from their own dam. The supplementary question ‘what they use instead’ was only answered by 2.8% of participants. The reason for this can only be hypothesized; it might be a possibility that farmers know they should keep a storage of frozen colostrum (82.6% stated they have one) and therefore answered that question accordingly when, in fact, they do not have a storage of frozen colostrum on their farm. The use of colostrum replacers is not very common in Austria, as shown by our results, probably because we have many organic producing farms for whom there is no permitted product available or because colostrum replacers, in general, are very expensive.

It is essential to test colostrum before feeding it to the calf, so dependent on the outcome of the test, the calf can be provided with a sufficient amount of immunoglobulins [51]. Our results show that only a quarter of farms (26.7%) in Austria test colostrum quality before feeding it to the calf. In a previous Austrian study, only 20.8% checked colostrum quality [26], so our results at least show an improvement. For colostrum testing, visual appraisal is wholly inadequate in predicting high (>50 g/L IgG) or low colostrum quality [52]. Other methods such as a colostrometer [53], brix-refractometer [54,55], or ColostroCheck^®^ (a cone-shaped device to rate the flow velocity of colostrum, ≥24 s = colostrum quality > 50 g IgG/L) [56] should be used. Although in the present study, more farms test colostrum quality, and fewer farms use visual appraisal for testing than in a previous study (20.8%, visual appraisal: 86.1% [26]), there is still room for improvement.

The time and quantity of first feeding colostrum to a calf is crucial. Calves fed with >2.5 L colostrum within the first three hours are less likely to suffer from FTPI [5]. Our study showed that many Austrian farmers do seem to be aware of this aspect of colostrum management. However, these two aspects of colostrum management alone are not sufficient to prevent FTPI in calves, as was shown in another study in Austria (Hartsleben et al., under review).

Besser et al. (1991) described different feeding methods and their likelihood for calves to develop FTPI. In the aforementioned study, 61.4% of calves nursed by the dam, 19.3% of calves fed by nipple bottle, and 10.8% of calves fed by esophageal tube were diagnosed with FTPI [57]. The use of an esophageal tube in general on every newborn calf is no standard practice in Austria. That might be because of the small farm structures and mostly family-owned farms where it is easy to care for each newborn calf individually. This fact could also be the reason why the most common answer was ‘colostrum is offered multiple times’ (65.5%). Our study showed that in foothills/flatland regions with larger farm structures, farms do use an esophageal tube within 2–6 h (23.1%) if the calf has not been drinking colostrum willingly by that time.

To the best of our knowledge, this was the first attempt to implement a scoring system to quantify herd and calf management practices. The scoring system might be used by farmers to assess the herd and calf management practices on their farms and to become aware of the areas that need to be improved. Additionally, standard operating procedures might be established, especially for small Austrian dairy farms. The findings of our study might be used by veterinarians in farm consultancies or other specialists and organizations providing advisory services to farms in order to reduce the risk of FTPI in calves.

## 5. Conclusions

The results of this study provide data on herd and calf management practices on dairy farms with a special focus on colostrum management practices in Austria. Furthermore, significant differences could be determined between alpine (west) and foothills/flatland (east) regions, organic and conventional farms, and full-time or part-time operated farms. In summary, eastern regions (foothills/flatland region: Burgenland, Styria, Upper and Lower Austria), conventional producing and full-time operated farms received higher scores than western regions (alpine regions: Vorarlberg, Tyrol, Carinthia, and Salzburg) than organic producing and part-time operated farms.

## Figures and Tables

**Figure 1 animals-13-02758-f001:**
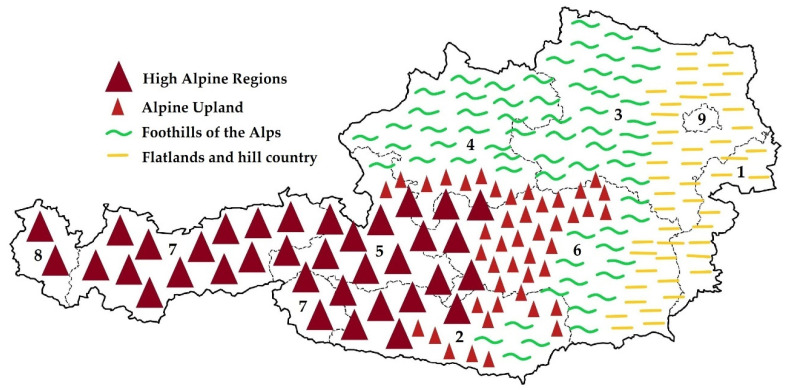
Schematic map of Austria’s geological structure and the nine federal states. Vienna is the capital city of Austria but is also considered a federal state. (1 = Burgenland, 2 = Carinthia, 3 = Lower Austria, 4 = Upper Austria, 5 = Salzburg, 6 = Styria, 7 = Tyrol, 8 = Vorarlberg, 9 = Vienna).

**Figure 2 animals-13-02758-f002:**
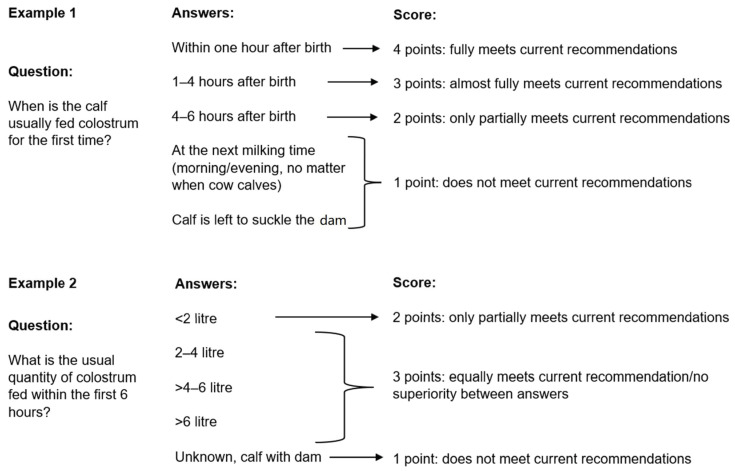
The figure shows the allocation of points based on two example questions.

**Figure 3 animals-13-02758-f003:**
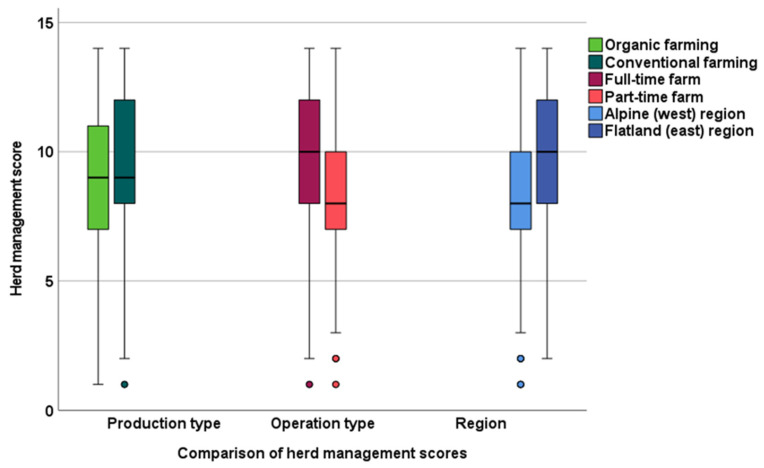
Comparisons on herd management scores in the alpine (west) and foothills/flatland (east) region of Austria, organic and conventional production types, and part-time and full-time operated farms.

**Figure 4 animals-13-02758-f004:**
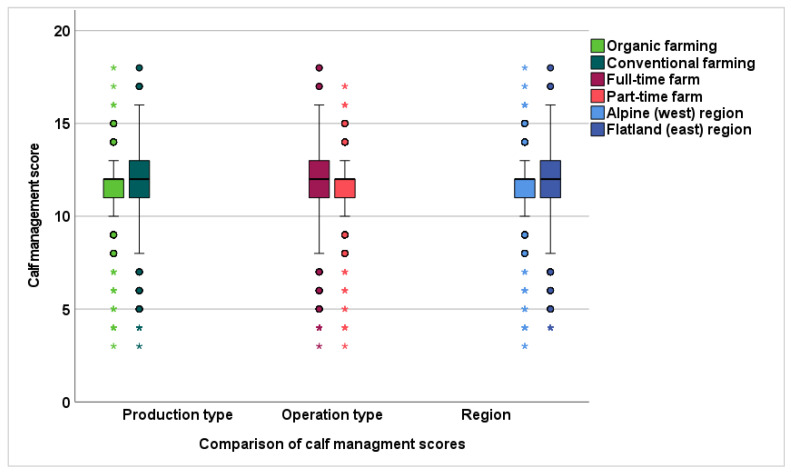
Comparisons on calf management scores in the alpine (west) and foothills/flatland (east) region of Austria, organic and conventional production types, and part-time and full-time operated farms.

**Table 1 animals-13-02758-t001:** The table gives an overview of the response rate (n = 2210) per federal state and the data from the official institutions/databases AgrarMarkt Austria (AMA) and VIS (Verbrauchergesundheitsinformationssystem). ^1^ Data extracted from AMA. ^2^ Data extracted from VIS. * Results from the underlying survey (n = 2210). BGL = Burgenland, CAR = Carinthia, LOAT = Lower Austria, UPAT = Upper Austria, SBG = Salzburg, STY = Styria, T = Tyrol, VBG = Vorarlberg.

Farm Structure	Federal State								Total Austria
	BGL	CAR	LOAT	UPAT	SBG	STY	T	VBG	
n ^1^ dairy cattle farmers per federal state (%)	76 (0.3)	1651 (6.9)	4012 (16.8)	6084 (25.5)	3270 (13.7)	3753 (15.7)	3879 (16.3)	1143 (4.8)	23,868 (100.0)
n * dairy cattle farmers responses (%)	8 (0.4)	151 (6.8)	404 (18.3)	451 (20.4)	296 (13.4)	341 (15.4)	439 (19.9)	120 (5.4)	2210 (100.0)
n ^2^ organic dairy cattle farmers per federal state (%)	7 (9.2)	373 (22.6)	839 (20.9)	1081 (17.8)	2017 (61.7)	993 (26.5)	1086 (28.0)	163 (14.3)	6559 (27.5)
n * organic dairy cattle farmers responses (%)	0 (0.0)	51 (33.8)	93 (23.0)	91 (20.2)	196 (66.2)	100 (29.3)	126 (28.7)	21 (17.5)	678 (30.7)
n ^2^ TGD membership per federal state (%)	69 (90.8)	1565 (94.8)	3848 (95.9)	5503 (90.5)	2575 (78.8)	3409 (90.8)	3738 (96.4)	1146 (100.0)	21,853 (91.5)
n * TGD members (%)	8 (100.0)	148 (98.0)	398 (98.5)	436 (96.7)	278 (93.9)	325 (95.3)	425 (96.8)	120 (100.0)	2138 (96.7)
n * part-time operated dairy cattle farmers (%)	0 (0.0)	50 (33.1)	88 (21.8)	117 (25.9)	147 (49.7)	101 (29.6)	271 (61.7)	60 (50.0)	835 (37.7)
n * full-time operated dairy cattle farmers answers (%)	8 (100.0)	101 (66.9)	316 (78.2)	334 (74.1)	149 (50.3)	240 (70.4)	168 (38.3)	60 (50.0)	1376 (62.3)

**Table 2 animals-13-02758-t002:** Survey results on general farm characteristics of the 2210 included dairy farms. The alpine region (West) shows the summary of answers from Vorarlberg, Tyrol, Salzburg and Carinthia. The foothills/flatlands region (East) shows the summary of answers from Burgenland, Styria, Upper and Lower Austria. Additionally, the production types (ORG = organic; CON = conventional) and the operating types (Part = part-time farming; Full = Full-time farming) are shown. N.A. = no answer, OLA = outdoor loafing area. * The breed category “brown Swiss” also included the “original brown Swiss” (Original Braunvieh).

Question	Answer Category	Region		Production Type		Operation Type		n (%) Total
		n (%) West	n (%) East	n (%) ORG	n (%) CON	n (%) Part	n (%) Full	
Farm size	≤10	314 (31.4)	69 (5.7)	136 (20.1)	247 (16.2)	325 (39.2)	58 (4.2)	383 (17.4)
	11–20	348 (34.8)	284 (23.6)	261 (38.6)	371 (24.3)	320 (38.6)	312 (22.7)	632 (28.7)
	21–30	172 (17.2)	335 (27.9)	155 (22.9)	352 (23.1)	133 (16.0)	374 (27.2)	507 (23.0)
	31–40	84 (8.4)	206 (17.1)	74 (10.9)	216 (14.2)	35 (4.2)	255 (18.6)	290 (13.2)
	≥41	83 (8.3)	308 (25.6)	51 (7.5)	340 (22.3)	17 (2.0)	374 (27.2)	391 (17.7)
	Total	1001	1.202	677	1.526	830	1373	2203
Breed	Simmental	592 (59.1)	1044 (86.9)	523 (77.5)	1113 (72.8)	554 (66.7)	1082 (78.8)	1636 (74.3)
	Holstein-Friesian	90 (9.0)	66 (5.5)	33 (4.9)	123 (8.0)	37 (4.5)	119 (8.7)	156 (7.1)
	Brown Swiss *	205 (20.5)	82 (6.8)	62 (9.2)	225 (14.7)	152 (18.3)	135 (9.8)	287 (13.0)
	Pinzgauer	61 (6.1)	4 (0.3)	37 (5.5)	28 (1.8)	40 (4.8)	25 (1.8)	65 (3.0)
	Tyrolean Grey	34 (3.4)	3 (0.2)	10 (1.5)	27 (1.8)	34 (4.1)	3 (0.2)	37 (1.7)
	Others and Crossbreeds	19 (1.9)	3 (0.2)	10 (1.5)	12 (0.8)	13 (1.6)	9 (0.7)	22 (1.0)
	Total	1001	1202	675	1.528	830	1373	2203
Average milk yield per cow per year in liter	N.A.	61 (6.1)	53 (4.4)	29 (4.3)	85 (5.5)	57 (6.8)	57 (4.1)	114 (5.2)
	2000–6500	309 (30.7)	185 (15.4)	273 (40.3)	221 (14.4)	271 (32.5)	223 (16.2)	494 (22.4)
	6501–7500	267 (26.5)	264 (21.9)	234 (34.5)	297 (19.4)	218 (26.1)	313 (22.7)	531 (24.0)
	7501–8700	218 (21.7)	305 (25.3)	108 (15.9)	415 (27.1)	173 (20.7)	350 (25.4)	523 (23.7)
	8701–14,000	151 (15.0)	397 (33.0)	34 (5.0)	514 (33.6)	115 (13.8)	433 (31.5)	548 (24.8)
	Total	1006	1204	678	1532	834	1376	2210
Housing type for lactating and dry cows	Freestall barn with OLA/pasture	386 (38.7)	507 (42.1)	427 (63.2)	466 (30.6)	258 (31.2)	635 (46.2)	893 (40.6)
	Freestall barn without OLA/pasture	52 (5.2)	451 (37.5)	9 (1.3)	494 (32.4)	88 (10.6)	415 (30.2)	503 (22.9)
	Tie stalls with OLA/pasture	541 (54.2)	202 (16.8)	238 (35.2)	505 (33.1)	446 (53.9)	297 (21.6)	743 (33.8)
	Tie stalls without OLA/pasture	19 (1.9)	43 (3.6)	2 (0.3)	60 (3.9)	36 (4.3)	26 (1.9)	62 (2.8)
	Total	998	1203	676	1.525	828	1373	2201

**Table 3 animals-13-02758-t003:** Survey results on herd-management practices part 1 with a special focus on herd-level colostrum management practices. The alpine region (West) shows the summary of answers from Vorarlberg, Tyrol, Salzburg and Carinthia. The foothills/flatlands region (East) shows the summary of answers from Burgenland, Styria, Upper and Lower Austria. Additionally, the production types (ORG = organic; CON = conventional) and the operating types (Part = part-time farming; Full = Full-time farming) are shown. ^1^ nested question to the previous question.

Question	Answer Category	Region		Production Type		Operation Type		n (%) Total
		n (%) West	n (%) East	n (%) ORG	n (%) CON	n (%) Part	n (%) Full	
Availability of a separate calving area	Yes	429 (42.8)	841 (70.0)	394 (58.3)	876 (57.3)	332 (40.0)	938 (68.3)	1270 (57.6)
	No	574 (57.2)	360 (30.0)	282 (41.7)	652 (42.7)	499 (60.0)	435 (31.7)	934 (42.4)
	Total	1003	1201	676	1.528	831	1.373	2.204
^1^ Cows actually calving in the separate calving area in %	All (100%)	110 (25.5)	226 (26.9)	78 (19.7)	258 (29.5)	84 (25.1)	252 (26.9)	336 (26.4)
	Almost all (90%)	176 (40.8)	376 (44.7)	162 (40.9)	390 (44.5)	136 (40.7)	416 (44.3)	552 (43.4)
	The majority (75%)	70 (16.2)	122 (14.5)	75 (18.9)	117 (13.4)	51 (15.3)	141 (15.0)	192 (15.1)
	Half (50%)	32 (7.4)	64 (7.6)	33 (8.3)	63 (7.2)	28 (8.4)	68 (7.2)	96 (7.5)
	Less than half (<50%)	43 (10.0)	53 (6.3)	48 (12.1)	48 (5.5)	35 (10.5)	61 (6.5)	96 (7.5)
	Total	431	841	396	876	334	938	1.272
Colostrum harvesting method	Milking machine	600 (59.9)	725 (60.6)	399 (59.3)	926 (60.7)	531 (63.9)	794 (58.1)	1325 (60.3)
	By hand	365 (36.5)	437 (36.5)	235 (34.9)	567 (37.2)	277 (33.3)	525 (38.4)	802 (36.5)
	Calf stays with dam	36 (3.6)	35 (2.9)	39 (5.8)	32 (2.1)	23 (2.8)	48 (3.5)	71 (3.2)
	Total	1001	1197	673	1.525	831	1367	2198
Availability of frozen colostrum stocks	Yes	763 (76.3)	1053 (87.9)	550 (81.2)	1266 (83.2)	637 (76.7)	1179 (86.2)	1816 (82.6)
	No	237 (23.7)	145 (12.1)	127 (18.8)	255 (16.8)	193 (23.3)	189 (13.8)	382 (17.4)
	Total	1000	1198	677	1521	830	1368	2198

**Table 4 animals-13-02758-t004:** Survey results on herd-management practices part 2 with a special focus on herd-level colostrum management practices. The alpine region (West) shows the summary of answers from Vorarlberg, Tyrol, Salzburg and Carinthia. The foothills/flatlands region (East) shows the summary of answers from Burgenland, Styria, Upper and Lower Austria. Additionally, the production types (ORG = organic; CON = conventional) and the operating types (Part = part-time farming; Full = Full-time farming) are shown. ^1^ nested question to the previous question.

Question	Answer Category	Region		Production Type		Operation Type		
		n (%) West	n (%) East	n (%) ORG	n (%) CON	n (%) Part	n (%) Full	n (%) Total
Time to calf/dam separation	Not at all (<20 min)	542 (54.3)	724 (69.3)	318 (47.1)	948 (62.2)	516 (62.2)	750 (54.7)	1266 (57.5)
	Up to 1 h	199 (19.9)	192 (16.0)	126 (18.7)	265 (17.4)	168 (20.2)	223 (16.3)	391 (17.8)
	1–4 h	101 (10.1)	147 (12.2)	79 (11.7)	169 (11.1)	67 (8.1)	181 (13.2)	248 (11.3)
	>4 h up to 1 Day	92 (9.2)	96 (8.0)	83 (12.3)	105 (6.9)	58 (7.0)	130 (9.5)	188 (8.5)
	>1 Day	51 (5.1)	27 (2.2)	43 (6.4)	35 (2.3)	15 (1.8)	63 (4.6)	78 (3.5)
	Nurse cow-calf rearing	2 (0.2)	4 (0.3)	6 (0.9)	0 (0.0)	1 (0.1)	5 (0.4)	6 (0.3)
	Dam-bound calf rearing	12 (1.2)	11 (0.9)	20 (3.0)	3 (0.2)	5 (0.6)	18 (1.3)	23 (1.1)
	Total	999	1.201	675	1.525	830	1370	2200
Colostrum harvest after parturition	Within 1 h	549 (55.1)	667 (55.4)	327 (48.4)	889 (58.3)	447 (53.9)	769 (56.1)	1216 (55.3)
	1–6 h	346 (34.7)	375 (31.2)	234 (34.6)	487 (32.0)	307 (37.0)	414 (30.2)	721 (32.8)
	Next milking time	84 (8.4)	144 (12.0)	96 (14.2)	132 (8.7)	67 (8.1)	161 (11.7)	228 (10.4)
	Calf stays with dam	18 (1.8)	17 (1.4)	19 (2.8)	16 (1.0)	8 (1.0)	27 (2.0)	35 (1.6)
	Total	997	1.203	676	1.524	829	1.371	2200
Udder cleaning before colostrum milking	Yes	831 (83.4)	1069 (88.9)	583 (86.2)	1317 (86.4)	717 (86.6)	1183 (86.2)	1900 (86.4)
	No	166 (16.6)	134 (11.1)	93 (13.8)	207 (13.6)	111 (13.4)	189 (13.8)	300 (13.6)
	Total	997	1203	676	1524	828	1.372	2.200
^1^ Udder cleaning methods	Wood wool	163 (19.7)	219 (20.6)	169 (29.1)	213 (16.3)	137 (19.2)	245 (20.8)	382 (20.2)
	Udder cloth wet	281 (34.0)	337 (31.7)	164 (28.2)	454 (34.7)	250 (35.1)	368 (31.3)	618 (32.7)
	Udder cloth dry	370 (44.7)	488 (46.0)	243 (41.8)	615 (47.0)	322 (45.2)	536 (45.6)	858 (45.4)
	Automatic (robotic system)	13 (1.6)	18 (1.7)	5 (0.9)	26 (2.0)	4 (0.6)	27 (2.3)	31 (1.6)
	Total	827	1062	581	1308	713	1176	1889

**Table 5 animals-13-02758-t005:** Survey results on calf management practices part 1 with a special focus on calf-level colostrum management practices (source of colostrum and colostrum testing methods). The alpine region (West) shows the summary of answers from Vorarlberg, Tyrol, Salzburg and Carinthia. The foothills/flatlands region (East) shows the summary of answers from Burgenland, Styria, Upper and Lower Austria. Additionally, the production types (ORG = organic; CON = conventional) and the operating types (Part = part-time farming; Full = Full-time farming) are shown. ^1^ nested question to the previous question.

Question	Answer Category	Region		Production Type		Operation Type		n (%) Total
		n (%) West	n (%) East	n (%) ORG	n (%) CON	n (%) Part	n (%) Full	
Colostrum from mother	Yes, always	721 (72.0)	739 (61.5)	458 (67.8)	1002 (65.6)	602 (72.4)	858 (62.5)	1460 (66.3)
	Mostly	8 (0.8)	7 (0.6)	8 (1.2)	7 (0.5)	4 (0.5)	11 (0.8)	15 (0.7)
	Yes, if dam has good colostrum quality	223 (22.3)	394 (32.8)	150 (22.2)	467 (30.6)	196 (23.6)	421 (30.7)	617 (28.0)
	Calf stays with dam	44 (4.4)	42 (3.5)	55 (8.1)	31 (2.0)	25 (3.0)	61 (4.4)	86 (3.9)
	No, not always	5 (0.5)	20 (1.7)	5 (0.7)	20 (1.3)	4 (0.5)	21 (1.5)	25 (1.1)
	Total	1001	1202	676	1527	831	1372	2203
^1^ Colostrum source if not from mother	Frozen colostrum	23 (95.8)	36 (97.3)	17 (100.0)	42 (95.5)	14 (100.0)	45 (95.7)	59 (96.7)
	Colostrum replacer	1 (4.2)	1 (2.7)	0 (0.0)	2 (4.5)	0 (0.0)	2 (4.3)	2 (3.3)
	Total	24	37	17	42	14	47	61
Assessment of colostrum quality	Yes	272 (27.0)	318 (26.4)	162 (23.9)	428 (27.9)	219 (26.3)	371 (27.0)	590 (26.7)
	No	734 (73.0)	886 (73.6)	516 (76.1)	1104 (72.1)	615 (73.7)	1005 (73.0)	1620 (73.3)
	Total	1006	1204	678	1532	834	1376	2210
Colostrum quality assessment method	Colostrometer	23 (8.6)	28 (9.0)	12 (7.5)	39 (9.4)	15 (6.9)	36 (10.0)	51 (8.8)
	Refractometer	35 (13.1)	71 (22.8)	25 (75.5)	81 (19.4)	25 (11.5)	81 (22.4)	106 (18.3)
	Visually	190 (71.2)	178 (57.2)	112 (69.6)	256 (61.4)	162 (74.7)	206 (57.1)	368 (63.7)
	ColostroCheck	19 (7.1)	34 (10.9)	12 (7.5)	41 (9.8)	15 (6.9)	38 (10.5)	53 (9.2)
	Total	267	311	161	417	217	361	578

**Table 6 animals-13-02758-t006:** Survey results on calf management practice part 2 with a special focus on calf-level colostrum management practices (colostrum feeding time, quantity of colostrum delivered to the newborn, and feeding methods). The alpine region (West) shows the summary of answers from Vorarlberg, Tyrol, Salzburg, and Carinthia. The foothills/flatlands region (East) shows the summary of answers from Burgenland, Styria, Upper and Lower Austria. Additionally, the production types (ORG = organic; CON = conventional) and the operating types (Part = part-time farming; Full = Full-time farming) are shown. (p.n. = post natum).

Question	Answer Category	Federal State		Production Type		Operation Type		n (%) Total
		n (%) West	n (%) East	n (%) ORG	n (%) CON	n (%) Part	n (%) Full	
Time from parturition to colostrum feeding	Within 1 h p.n.	600 (60.2)	692 (57.8)	375 (55.6)	917 (60.3)	480 (58.0)	812 (59.4)	1292 (58.9)
	1–4 h p.n.	296 (29.7)	366 (30.6)	201 (29.8)	461 (30.3)	265 (32.0)	397 (29.0)	662 (30.2)
	4–6 h p.n.	27 (2.7)	23 (1.9)	16 (2.4)	34 (2.2)	24 (2.9)	26 (1.9)	50 (2.3)
	Next standard milking time	43 (4.3)	77 (6.4)	38 (5.6)	82 (5.4)	40 (4.8)	80 (5.9)	120 (5.5)
	Calf suckles the dam	31 (3.1)	40 (3.3)	44 (6.5)	27 (1.8)	19 (2.3)	52 (3.8)	71 (3.2)
	Total	997	1198	674	1521	828	1367	2195
Quantity of colostrum fed within the first 6 h after birth	<2 L	182 (18.2)	230 (19.1)	94 (13.9)	318 (20.8)	177 (21.3)	235 (17.1)	412 (18.7)
	2–4 L	680 (67.9)	868 (72.2)	486 (72.0)	1062 (69.5)	556 (66.8)	992 (72.3)	1548 (70.2)
	>4–6 L	101 (10.1)	66 (5.5)	51 (7.6)	116 (7.6)	77 (9.3)	90 (6.6)	167 (7.6)
	>6 L	11 (1.1)	5 (0.4)	4 (0.6)	12 (0.8)	9 (1.1)	7 (0.5)	16 (0.7)
	Unknown, calf with dam	27 (2.7)	34 (2.8)	40 (5.9)	21 (1.4)	13 (1.6)	48 (3.5)	61 (2.8)
	Total	1001	1203	675	1529	832	1372	2204
Colostrum feeding equipment	Bucket	524 (52.4)	366 (30.6)	304 (45.0)	586 (38.5)	353 (42.5)	537 (39.3)	890 (40.5)
	Nipple bottle	412 (41.1)	788 (65.8)	307 (45.4)	893 (58.7)	434 (52.3)	766 (56.0)	1200 (54.6)
	Esophageal tube	14 (1.4)	1 (0.1)	5 (0.7)	10 (0.7)	14 (1.7)	1 (0.1)	15 (0.7)
	Calf stays with dam	50 (5.0)	42 (3.5)	60 (8.9)	32 (2.1)	29 (3.5)	63 (4.6)	92 (4.2)
	Total	1000	1197	676	1521	830	1367	2197
Calves not drinking well receive colostrum	Immediately esophageal tube	80 (8.0)	153 (12.8)	44 (6.5)	189 (12.5)	58 (7.0)	175 (12.8)	233 (10.6)
	Esophageal tube within 2–6 h	186 (18.7)	277 (23.1)	127 (18.8)	336 (22.2)	155 (18.8)	308 (22.5)	463 (21.1)
	By esophageal tube, in general	18 (1.8)	2 (0.2)	6 (0.9)	14 (0.9)	16 (1.9)	4 (0.3)	20 (0.9)
	Colostrum offered multiple times, in no case esophageal tube	694 (69.8)	742 (61.9)	491 (72.5)	945 (62.4)	583 (70.7)	853 (62.4)	1436 (65.5)
	Colostrum offered multiple times, later esophageal tube	15 (1.5)	22 (1.8)	8 (1.2)	29 (1.9)	11 (1.3)	26 (1.9)	37 (1.7)
	Others (vet, supplements, etc.)	1 (0.1)	2 (0.2)	1 (0.1)	2 (0.1)	2 (0.2)	1 (0.1)	3 (0.1)
	Total	994	1198	677	1515	825	1367	2192

## Data Availability

Data are available within the article or in its Appendix A.

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
