# Peer review of "Assessment of Herd, Calf, and Colostrum Management Practices on Austrian Dairy Farms Using a Scoring System"

_animals, 2023, doi:10.3390/ani13172758_

Round 1
Reviewer 1 Report
Review Animals August 2023
Abstract and summary: I would question the external validity of these results since they are very particular to one country’s farming systems. Also the response rate is reasonably low 14.3% which may also invalidate the conclusions from the work.
The allocation of scoring systems seems arbitrary and more explanation is needed. In fact this manuscript would be better presented as a descriptive piece of work without allocating an arbitrary and misleading scoring system.
Introduction section
Line 55: ‘The amount of maternal colostrum needed is depending on the concentration of immunoglobulins, the ingested volume of colostrum and the ability of the calf’s gut to absorb the immunoglobulins (apparent efficiency of immunoglobulin absorption).’
Suggest rewording ‘The amount of maternal colostrum needed depends on the concentration of immunoglobulins, the ingested volume of colostrum and the ability of the calf’s gut to absorb the immunoglobulins (apparent efficiency of immunoglobulin absorption).
In fact this should be modified to include the timing of administration of colostrum (which will affect AEA) and you should add optimal volumes and timings here.
Line 67-72 Please expand on the mechanisms by which bacterial species (in particular coliform species) may prohibit IgG absorption as there are many mechanisms by which this is proposed to occur- and add references – Godden etc.
Line 74-78 there is also some evidence of poorer first lactation milk yields and prolonged time to first service conception which should be added and referenced here.
Line 86 ‘Aforementioned’ not ‘forementioned’
Lines 87-105 It may be useful to an international readership to include details on the fate of milk produced by these small part time farms- is it mostly for cheese? Is it mostly for local consumption? Also are the farms all year round calving systems? Are they pasture based? Explain the systems in more detail to ‘paint the picture’ for international dairy vets and farmers.
Line 102 was there a specific hypothesis that certain farming systems would be inferior to others. It is understood that this is a descriptive study but perhaps a more weighty hypothesis would be useful.
Materials and methods section
Did you beta test the survey questions on anyone? This would be deemed best practice! How were the survey questions selected? Did you conduct a literature review? What expert opinion did you seek? Would it be possible to supply the supplementary material survey in English rather than in German?
Section 2.3 line 138 why did you only implement a scoring system for 13 questions when there were 24 questions in total.
Line 141 I don’t understand this I’m afraid- if the maximum number of points is 4 for each question and there are 13 questions (how these were selected from the 24 is still a mystery!) then the total possible maximum is 4 * 13 which is not equal to 32??
Line 145 which peer reviewed journals?? How did you conduct your searches? What were your criteria?
I think that by allocating answers as ‘correct’ and ‘incorrect’ you inherently bias your findings as in many cases there is no ‘correct’ answer and no ‘one size fits all’ approach (as evidenced by your own admission that in some cases it was simply not possible to classify answers as ‘correct’ or ‘incorrect’. You may in fact be able to achieve excellent passive immunity even if you do not do everything ‘correctly’, for example, you do not harvest your colostrum using a milking machine. Since this is a descriptive study I am not sure that the allocation of points is entirely necessary to the hypothesis.
Figure 2 in example 2 there is only a possibility of 3 maximum points which is at loggerheads with the description of the point allocation system above.
Section 2.4 you need to explicitly explain why you needed to use two separate external databases to populate this information.
Line 179: you do not state in the results section how many responses needed to be removed for ‘implausibility’ or ‘contradiction’ and what these were.
Line 193-194 presumably these descriptive statistics were calculated prior to categorisation of the variables so perhaps describe this before you describe how variables were catergorised.
Line 197 to 202 since you have categorised many of your variables I would suggest Chi squared analysis and the Cochran Armitage test for trend (to maximise power for multiple comparisons) to investigate biologically plausible relationships between the variables
Results section
In Table 1 I am unsure of the relevance of where the data came from (either external database or survey) and I think it overcomplicates the table so would suggest removing. Decide which is most reliable measure and use this.
Line 225 suggest rewording: ‘General farm characteristics were split…’ as the rest of the words here are superfluous.
Line 226 to 250 all of this information could (and should) be included in Table 2 rather than in the text. It is messy and a bit hard to follow. It would also be useful for international readers to understand which regions are ‘East’ and which are ‘West’ by your classification for this work.
In all tables please use ‘n’ not ‘N’ for the number of observations.
In Table 2 remove the Total row as in every case this will be 100%
Table 3 top right hand corner why is there a ‘9’?? Should this say ‘question’?
In Table 3 ‘Cows actually calving in the separate calving area in %’ the convention is to write ‘the majority’ not ‘the most’ but to be honest you could just include the percentages if this is what you defined to the survey respondents.
Again in all Tables you could remove the Total rows at the bottom of each subsection.
It would be useful to indicate in your tables the ‘nested’ questions, for example the respondents can only say which colostrum quality assessment they use if their response to the previous question is ‘yes’.
3.4.1 and 3.4.2 this information is already included in the Tables with the regions categorised as ‘East’ and ‘West’. Presumably the authors deem this categorisation as most relevant which makes me question the need to describe each region individually in the prose. If the detail of each specific region is indeed important I would include this in the tables (as also suggested lines 226-250).
Please remove Figures 3 and 4 as they are of limited usefulness- many overlapping confidence intervals and really no difference.
Lines 304-349 I have already indicated that I think this scoring system is intrinsically flawed and misleading and so I would remove all of this.
Discussion section
I would suggest focussing only on the interesting and relevant information here and putting your work in the context of others. For example: lines 497-499 the number of farms who test colostrum in Austria is approximately 26.7% according to your survey but how does this compare to other work? Please expand this suggestion and include comparison of your work throughout the discussion section as for now this section is just statement of fact with no real discussion.
Line 352 repetition from results section – you have already reported the response rate.
Line 368 the farmers here may not necessarily be more ‘educated’ but perhaps more ‘motivated’?? Consider rewording.
Line 369-375 questionable relevance suggest removing.
Section 4.2 It is acknowledged that some of the limitations of the ‘scoring system’ are listed here but it still needs to be removed as it is biased and misleading. This work would be much better presented as a simple descriptive survey without this somewhat arbitrary scoring system.
Section 4.3 line 408: repetition, you have already reported the mean herd size in Austria. Would it be more appropriate to emphasise the foothills or alpine regions instead of categorising the regions as ‘East’ and ‘West’ as you have done in your tables?
Section 4.4 line 429 you could expand on this idea- why would calves left with their dams be more likely to suffer from ftpi?
Line 450 ‘calve’ not ‘calf’
Line 462 ‘have frozen colostrum’ remove ‘a’
Line 464 -468: ‘We assumed that when colostrum is harvested by hand the udder is not sufficiently cleaned beforehand, but only 13.6 % explicitly stated that they do not clean the udder before milking. So, it can be presumed that some of those farms that do harvest colostrum by hand also clean the udder beforehand, although we did not ask if they did so by using gloves.’ Why did you assume that they would not clean the udder before colostrum harvest? If the did clean the udder they would not do so by ‘using gloves’ they would use a spray or dip please reword as while gloves may reduce contamination from the milkers hands they are not a cleaning method.
Line 480 the start of the ‘calf management’ section continues to discuss ‘colostrum management’ here by discussion pooling. Please consider reworking your subheadings to be more appropriate.
Line 483 ‘not’ not ‘no’
Line 483 ‘Due to small farm sizes, many times there is only one cow in parturition at a time.’ Suggest reword ‘on small farms there is often only one cow calving at any one time’
Line 503 ‘visual appraisal’ is wholly inadequate and you should state and reference this.
Line 507 ‘showed’ not ‘shows’ please write in the past tense.
Line 533 ‘materials’ is misspelled.
The discussion section needs to be much more focussed with clearer subheadings and only the salient points in context of the international literature.
Only minor edits needed mostly detailed in line by line review.
Author Response
Dear Reviewer,
Thank you for reviewing our manuscript.
Please find the detailed response to your comments in the attached word document.
Yours sincerely,
Katharina Lichtmannsperger (on behalf of the authors)

Reviewer 2 Report
The manuscript of Hechenberger and coworkers provides an overview of calf management practices in several regions of Austria based on an online survey using a scoring system. This is an interesting piece of research mainly because of the wide distribution of the survey and the relatively high number of responders. This allows for getting an overview of the situation in Austria as an example for a region with a small-scaled dairy industry.
The study is well performed and the manuscript is well written. The introduction is an easy read and leads to the objective of the study although some shortening of the first paragraph (L55-61) would be beneficial. The study design in clearly described, some indications for improvement of the statistical analysis are given below. The results section is very long and crammed with details and numbers which impedes reading and understanding. This should be reorganized. The discussion is broad and reflects current knowledge. Some parts would better fit into the results section (e.g., L354-357 and others) or the M+M section (e.g., 477-479), please review. The conclusional remarks are not completely based on the own results and should be reconsidered. Mainly L 523-524, 527-528, 530-532 better fit as a last paragraph of the discussion. The supplementary material is very detailed. From my perspective, the original questionnaire in German is dispensable because all questions including the answers and the scoring are given in table S2 (numbering of questions is not consistent between Tables S1 and S2).
With respect to an adequate data analysis, I really miss a multivariable model with the variables production type, operation type and region. In your discussion you are discussing possible interactions (e.g., L394-406). A multivariable model might elucidate the interactions, random effects, and covariances between production type, operation type and region. Please test if the variables are approximately normally distributed (otherwise a transformation would be helpful) and provide a multivariable model.
Some suggestions to improve the organization of the manuscript:
- All tables contain a lot of numbers and they are not easy to read. Please improve readability by using two lines for numbers and percentages in all cases.
- Section 3.2 is rather long and contains a lot of information. Please use paragraphs to divide the aspects and reconsider if it is necessary to give all the details about for instance breed and milk yield in the single federal states.
- Table 2-5: ‘100%’ in the line Total is dispensable.
- Reconsider if Table 2-5 would better fit into the supplemental material and Figure S1 better fit into the section 3.5.1 of the main manuscript (preferably with figure titles ‘herd management’ and ‘calf management’).
Please shorten the abstract to a total of about 200 words maximum and place the question addressed in a broader context (see chapter manuscript preparation on the mdpi website).
Please revise the format of your reference list and use the journals formatting (e.g., abbreviated journal name).
Minor comments
L 157 References should be numbered according to their first appearance in the manuscript.
L 161 calf is left to suckle the dam
L 310 Figure legend: Alpine (west) region
In general, this manuscript fits thematically within the scope of the special issue of Animals journal and it should be published in this special issue. I sincerely hope that my suggestions will enhance this manuscript. However, if I have made any errors or misinterpretations, I apologize in advance.
Author Response

(The authors gave the same response as above.)

Reviewer 3 Report
Title: Assessment of herd, calf and colostrum management practices on Austrian dairy farms using a scoring system
The objectives of the study were to describe herd, calf and colostrum management on Austrian dairy farms and to explore differences between regions (alpine/flatlands), organic and conventional producing farms and full-time or part-time operated farms. In conclusion, farms in foothills/flatland zones, traditional producing farms, and full-time-run farms obtained higher herd management rankings. Despite the fact that the current work is well conceived and produces fascinating results, server concerns were discovered throughout the paper and must be resolved by the authors before further evaluation. Please check the comments below for further information.
Abstract:
L23: Please indicate the number of treatments and experimental design utilized in this study.
Introduction
L65: How often is Failure of Transfer of Passive Immunity (FTPI) in newborn calves?
The author has prepared an intriguing introduction that is relevant to the issue. As a result, there are no changes in this area.
Materials and methods
L165-175: Two major Austrian databases were employed in the investigation to collect data on farm structure, production type, and Animal Health Service membership. Will this have an impact on the data analysis?
L177-206: How is the design and planning of the experiment to avoid dislocation?
L177-206: Diet is another major component influencing the immune system. Are the foods utilized in this research the same? and whether or whether the experimental work will be accurate?
Results
L225-250: Each component of the trial's outcomes should have its own written description. To make it easier for readers to grasp.
L259-264: In this area, the author should provide further explanations.
L330-340: Why do the authors report differing Herd management score and calf management score values (percentiles and median)?
Discussion
L418: Why was the average milk yield lower in alpine regions compared to foothills/flatland regions? Please describe the mechanism.
Conclusion
L522: The experiment looks to be a large survey research. So, is it feasible to propose a treatment to lessen the resolution of FTPI sickness in calves?
Minor editing of English language required.
Author Response

(The authors gave the same response as above.)

Reviewer 4 Report
The majority of my concerns are in regard to English grammar/wording.
Line 14 --- use "born" not "borne"
Line 16 --- it is true we want the colostrum we use to be > 50 g/L of IgG. However, what is the total amount the calf should consume? This should be added to this simple summary.
Line 56 --- use "dependent" rather than "depending"
Line 72 --- use "collecting" rather than "milking"
Line 94 --- Remove "Especially"
Line 167 --- use "established" rather than "installed"
Line 175 --- use "required" rather than "forced"
For most of the Tables, formulate them so that all the data within a row are level. For example, on table 1, the first row under BGL, 76 is on top of (0.3) versus CAR where 1651 is even with (6.9). If they data is level, it will make reading the data easier.
Lines 389-390 --- Also, in the future, if you could have actual data on Passive Transfer from these herds, I believe it would make the scoring system even stronger.
Lines 391-392 --- If the scores between the regions are not statistical, than it should be noted at this point. Are the differences simply numerically different (p>0.10), trending (p= 0.05-0.10), or statistically (p<0.05)? This needs to be clarified before publishing.
Line 431 --- End the sentence "This might be due to the philosophy of organic producing farms." Start a new sentence with, "Although, it is indicated by studies that calves have less FPTI when separated from the dam within 3 hours.[29,30]
Line 459 --- reword to "Results from our study show that in the foothills/flatland regions the information on storing frozen colostrum is more well-known than in alpine regions, see table 3.
Line 483 --- reword to "In Austria, it is not a common practice to pool colostrum.
The English is good and only minor revisions are necessary.
Author Response

(The authors gave the same response as above.)

Round 2
Reviewer 2 Report
The authors adressed all comments and improved the quality of the manuscript.
Reviewer 3 Report
Congratulation​s!